# Identification of a Green Algal Strain Collected from the Sarno River Mouth (Gulf of Naples, Italy) and Its Exploitation for Heavy Metal Remediation

**DOI:** 10.3390/microorganisms10122445

**Published:** 2022-12-10

**Authors:** Lucia Barra, Angela Sardo, Maria Moros, Arianna Smerilli, Pasquale Chiaiese, Isabella Percopo, Elena Cavalletti, Christian Castro-Hinojosa, Sergio Balzano

**Affiliations:** 1Department of Ecosustainable Marine Biotechnology, Stazione Zoologica Anton Dohrn, Contrada Torre Spaccata, 87071 Amendolara, Italy; 2Department of Ecosustainable Marine Biotechnology, Stazione Zoologica Anton Dohrn, Via Acton 55, 80133 Naples, Italy; 3Institute of Nanosciences and Materials of Aragon, C/Mariano Esquillor 15, 50018 Zaragoza, Spain; 4Department of Agricultural Sciences, University of Naples Federico II, Via Università 100, 80055 Portici, Italy; 5Department of Research Infrastructures for Marine Biological Resources, Stazione Zoologica Anton Dohrn, Villa Comunale 1, 80121 Naples, Italy; 6Department of Marine Microbiology and Biogeochemistry (MMB), Netherland Institute for Sea Research (NIOZ), Landsdiep 4, 1793 AB Texel, The Netherlands

**Keywords:** heavy metals, bioremediation, microalgae, isolation, taxonomical identification, growth rate reduction, heavy metal accumulation

## Abstract

Heavy metals (HMs) can induce both chronic and acute harmful effects on marine and freshwater biota. The environmental impact of HMs in freshwater, seawater, soil, and wastewater can be limited using microbes, including microalgae, that are able to remove metals from environmental matrices. Indeed, they can passively adsorb and actively accumulate these persistent pollutants within their organelles, limiting their detrimental effects on cellular metabolism. The Sarno River is a 30 km long freshwater stream located in Southern Italy, polluted by partially untreated municipal, agricultural, and industrial wastewaters. In spite of this, microalgal cultures from Sarno River or Sarno River Mouth have never been established. In the present study, we isolated a green algal strain from the Sarno River Mouth and determined its ability to grow in polluted seawater containing different concentrations of cadmium, lead, or zinc. This strain was found to be able to accumulate these elements within its biomass in a dose-dependent manner. Growth inhibition experiments confirm the relatively low toxicity of Cd and Pb below 50 µM, while algal growth was seriously affected in Zn-amended media. To the best of our knowledge, this is the first study focused on the ability of microalgae from Sarno River Mouth to tolerate and uptake HMs.

## 1. Introduction

Heavy metals (HMs) can induce both chronic and acute harmful effects on marine [1,2,3] and freshwater [4,5] biota. The presence of these contaminants in groundwater, freshwater (lakes, rivers, and streams), seawater, soil [6], and wastewater treatment plants, has been well documented in the literature [6,7,8]; microbes have been reported to inhabit HM-contaminated sediments [9], and can be used for the removal of HMs from environmental matrices [10,11,12,13,14,15]. In this context, microalgae are currently considered as suitable microorganisms for HM removal, since they are able to uptake HMs from a wide range of environments, including wastewaters and sediments, through mechanisms of adsorption and compartmentalization [11,12,16,17,18,19,20,21,22,23,24]. Since HMs can interfere with cellular metabolism, they pose serious threats to human and ecosystem health when their concentration in the environment exceeds specific thresholds [25,26]. Even though some HMs, such as copper and zinc, are essential for cellular metabolism [27,28], they can become detrimental at high concentrations [29,30,31]. Organisms can take up non-essential HMs using the same transport mechanisms that they typically use for essential HMs, including when the latter are present in the environment [32]. For example, the role of cadmium, usually hazardous for microorganisms, changes for the diatom *Thalassiosira weissflogii* under zinc-depleted conditions, becoming essential [33,34]. HMs are largely used in various industrial processes, such as mining, plating, smelting, as well as the manufacture of plastics, textiles, and varnishes, and oil refinery processes [35]. HM pollution of aquatic ecosystems is mostly driven by the discharge of partially untreated agricultural and industrial wastewaters into the environment [19]. HMs accumulate along the food chain, affecting different degrees of biological organization, such as cells, tissues, organs, and apparatuses, up to live stock and fishes used for human consumption. The process, named biomagnification, poses a health risk for humans [36,37,38]. HM-polluted environments are mostly located within densely populated areas as well as in close proximity to large industrial plants. The Sarno River is a 30 km long freshwater stream located in Southern Italy that flows into the Gulf of Naples. Improperly treated municipal, agricultural, and industrial wastewaters have been discharged directly into Sarno River, or its tributaries, for several decades [39]. In particular, discharge from the tanning and canning industries lead to significant pollution in terms of hydrocarbons and HMs along the river [40,41], as well as in the seawater basin in front of its mouth [8]. Microorganisms thriving within the Sarno River and Sarno River Mouth are likely to be adapted to survive and grow in the presence of high levels of HM pollution. They are likely to possess physiological and metabolic features of biotechnological interest for bioremediation purposes. Specifically, polluted waters can be partially treated by living microorganisms that can both passively adsorb and actively accumulate HMs. In particular, eukaryotic microbes can compartmentalize HMs within vacuoles, mitochondria, or plastids, limiting their detrimental effects on cellular metabolism [19,42]. Microalgae are suitable candidates for water treatment and bioremediation because of (1) their autotrophic behavior, implying that they do not require organic substrates for their growth, (2) their fast growth rates, (3) their relatively high biomass productivity [43,44], and (4) their high biosorption and metal internalization capabilities [12,18,42,45,46]. The present study was mainly aimed at determining the ability of a microalgal strain, isolated from Sarno River Mouth, to grow in artificially polluted seawater and to accumulate selected HMs within its biomass. Our results suggest that this novel species could be considered a suitable candidate to remove HMs from aquatic environments, especially non-essential metals such as Cd and Pb.

## 2. Materials and Methods

### 2.1. Sediment Sampling, Microalgae Isolation, and Culturing

Surface sediment was collected at 5 m depth from ca. 200 m off the Sarno River Mouth (40.72875° N, 14.466432° E) using a Van Veen grab sampler. A salinity of 30 was measured in the seawater overlaying the sediment collected. The sediment was stored at 4 °C until further processing. In the laboratory, 1 g of sediment was poured into a Petri dish pre-filled with 30 mL f/2 medium [47]. A 1:1 dilution of the culture medium was achieved with f/2 amended with 20 µM copper sulfate, 2 µM lead nitrate, 50 µM zinc sulfate, and 10 µM cadmium chloride, to stimulate the growth of the most HM-tolerant species. subsequently, 2 mM buffer solution (Tris-HCl, pH 7.8) was added to the medium to prevent substantial pH variations during algal growth. The resulting enrichments were incubated for 1 week at 20 °C. Artificial light intensity (ca. 100 μmol m^−2^ s^−1^) was provided by daylight fluorescent tubes with a 12:12 light: dark regime. The microalgal strain, named BS14, was isolated by the capillary pipette method under an Axiophot inverted light microscope (Zeiss, Germany), and then cultured in f/2 medium, prepared at salinity 30, at 18 °C under a 12:12 light: dark photoperiod. Cell growth was monitored by enumeration under a Zeiss Axiovert 200 light microscope (Carl Zeiss, Oberkochen, Germany) using a Sedgewick Rafter chamber.

### 2.2. Taxonomical Identification through 18S rRNA and rbcL Genes

Cells were harvested from cultures during the late exponential phase. Specifically, 50 mL of culture, corresponding to ca. 70,000,000 cells, was collected in falcon tubes and centrifuged for 15 min at 2900 rcf. DNA extraction and purification was carried out as described by Kooistra et al. [48] with some slight modifications: after the chloroform–isoamyl alcohol treatment, the aqueous phase was mixed with an equal volume of ice-cold isopropanol, then left at −20 °C for 1 h, and subsequently centrifuged in an Eppendorf microfuge at 16,000 rpm for 15 min. The DNA was subsequently cleaned, precipitated, and resuspended in a 3 M ammonium acetate solution. We amplified both the 18S rRNA and the *rbcL* genes from the genomic DNA extracted from BS14. PCR reactions were carried out using a C1000 Touch Thermal Cycler (Applied Biosystem, Monza, Italy) in a final volume of 30 μL containing 25–100 ng of DNA, 6 μL of 5× PCR Go-Flexi buffer (Promega, Madison, WI, USA), 3 μL of Mg^2+^ 25 mM, 0.6 μL of dNTPs (Roche, 10 mM for each nucleotide), 0.5–1 μM of primers, and 0.2 μL of 5UI/μL Taq (Promega). The primers used to amplify the 18S rRNA gene were 528F-mod [49] and 1818R [50], whereas the *rbcL* gene was amplified using primers *rbcL*168 and *rbcL*328 [51]. For the 18S rRNA gene, the PCR consisted of initial denaturation at 94 °C (5 min), then 34 cycles at 94 °C (1 min), 55 °C (1 min), and 72 °C (2 min), and a final extension at 72 °C (10 min). The temperature cycling for the *rbcL* gene was set as follows: 94 °C (5 min); 39 cycles of 94 °C (1 min), 50 °C (2 min), and 72 °C (3 min); and 72 °C (10 min) final extension. The amplified fragments were purified using a Nucleospin gel and PCR clean-up kit (Mackerey Nagel, Dueren, Germany) following the manufacturer instructions, and the final product was eluted in 20 μL. The purified PCR products were then sequenced on an Applied Biosystems 3730 DNA Analyzer with 48 capillaries using the BigDye^®^ Terminator v3.1 Cycle Sequencing kit (Life Technologies, Carlsbad, CA, USA). The resulting sequences were analyzed and aligned with reference sequences from GenBank using Bioedit software (https://bioedit.software.informer.com/7.2, (accessed on 2 March 2021) [52]. For both the 18S rRNA and the *rbcL* genes, *Tetraselmis* sequences obtained from GenBank were aligned with sequences of our strain using ClustalW [53], and phylogenetic trees were built using Mega software (Kumar et al. 2016). For the 18S rRNA gene, *Prasinoderma coloniale* AB058379 and *Mesostigma viridae* AJ250109 were used as outgroups for a total of 17 sequences. Highly variable positions were manually removed from the alignment, which finally included 1105 bp. For the *rbcL* gene, only four sequences were available on GenBank; these were used for the alignment, along with *Radiococcus polycoccus* JQ259914 (outgroup). The alignment included 308 unambiguously aligned positions. Phylogenetic relationships were explored using the maximum likelihood (ML) algorithm, and bootstrap values were estimated using 1000 replicates. 

### 2.3. Impact of HMs on Cell Growth and Morphology

The HM concentrations used in this experiment were selected after preliminary screening performed in 15 mL glass tubes, in which algal growth was evaluated by measuring the red fluorescence on a Turner design fluorometer (Sunnyvale Ca; data not shown). The HM concentrations used (Table 1) were chosen during the pre-screening process. The lowest concentrations were selected based on values commonly observed for the Sarno River Mouth [8], since they were found to not substantially impair microalgal growth. The highest concentrations (i.e., Cd-250, Pb-200, and Zn-250) corresponded, instead, to those at which cells were still able to survive, but showed very low or no growth.

For the experiment, we used a culture medium with the same composition as the f/2 medium except EDTA, which was not added (f/2-mod). HM salts were then added to the medium separately, as described in Table 1. The experiments were performed in triplicate using 75 cm^2^ vented cap sterile flasks (Falcon), filled with 100 mL f/2-mod and amended separately with different metals at different concentrations. A total of 30 flasks were incubated for 5 days at 18 °C under a 12:12 light: dark regime with 180 µmol photons m^−2^ s^−1^. Cell growth was monitored daily by microscopy counts, as described above, for 7 days. Cells were counted immediately after collection, or fixed with a 0.5% Lugol’s iodine solution and stored at 4 °C until microscopy counts. The initial cell density was ca. 200,000 cells mL^−1^. 

In order to assess the growth inhibition percentage for each condition, we initially determined the specific growth rates (*µ*), as described previously [54,55].

The growth inhibition percentage was then calculated as follows:
$$\%\;I\;=\frac{(\mu_{c}-\mu_{t})*100}{\mu_{c}}$$where *µ_c_* is the growth rate of the control culture, and *µ_t_* is the treated culture growth rate. 

Data are expressed as mean ± standard deviation (sd), and statistical analyses were performed by Shapiro–Wilk and Bartlett tests using the SPSS 27.0 software [56]. To assess the impact of cadmium, lead, and zinc on the morphology of BS14, cells exposed to the highest concentrations of HMs, as detailed below, were observed and photographed using a Zeiss Axiovert 200 light microscope (Oberkochen, Germany) equipped with Nomarski DIC, PH, and BF optics. In addition, cell surface parameters were measured from about 25–30 cells in each treatment, using the software Colibrì 5 (Zeiss, Germany); through the software, the cellular shape was transformed to an ellipse.

The statistical variation in cellular surface size, associated to different treatments, was determined through a *t*-test (GraphPad Prism 9).

### 2.4. Determination of HM Concentration in Microalgal Biomass

The HM content was determined in the microalgal biomass collected at the end of the experiment by inductively coupled plasma mass spectrometry (ICP-MS). HMs in microalgal biomass were determined for selected treatments only. Specifically, we selected cultures treated with low and sublethal doses of cadmium and lead, i.e., Cd-50 and Cd-250 as well as Pb-10 and Pb-50, for cadmium and lead, respectively. For zinc, only treatments amended with the lowest doses (20 µM) were able to grow; we thus only analyzed the Zn-20 treatment using ICP-MS. Cultures were transferred in 50 mL falcon tubes, and microalgal cells were then separated from the medium by centrifugation using a swing-out centrifuge (Allegra X12 R, Beckman Coulter Inc., Palo Alto, CA, USA) at 2500× *g*, for 10 min. To determine the HM content in microalgal biomass, wet pellets were transferred to pre-weighed 1.5 mL Eppendorf tubes, centrifuged again for 1 min to remove the residual supernatant, and were then dried in a freeze dryer (Edwards modulyo EF4, Burladingen, Germany) overnight. Lyophilized weight was determined by the difference between the gross weight and the tare weight. Pellets were digested in 180 µL nitric acid 65% *v*/*v*, and incubated at 80 °C for 90 min. Subsequently, they were treated with 180 µL hydrogen peroxide (33% *v*/*v*) at 80 °C until digestion was completed, and then diluted in demineralized water up to a final volume of 10 mL. HMs were analyzed using an optical emission spectrophotometer with inductively coupled plasma (Agilent 5100), coupled with a quartz Meinhardt concentric nebulizer and a Scott-type spray chamber with axial view for all the measurements. Calibration curves were generated from 0 to 100 ppb using certified reference materials (Merck, Darmstadt, Germany). The limits of detection for the metals under the analyzed conditions were 0.01 ppb for Cu, Cd, and Pb, and 6 ppb for Zn.

## 3. Results

### 3.1. Taxonomical Identification 

Phylogenetic analyses indicate that the 18S rRNA gene sequence of BS14 groups with two sequences from *Tetraselmis convolutae*, forming a moderately supported clade (54% bootstrap support) that clusters with *Tetraselmis* sp. KT860916, forming a well-supported clade (84% bootstrap), separated from other *Tetraselmis* species (Figure 1). The *rbcL* gene sequence of our strain also grouped with a sequence from *T. convolutae* with 90% bootstrap support (Appendix A). Current data thus strongly suggest that the strain BS14 belongs to *T. convolutae*.

### 3.2. Response to Heavy Metal Stress

*Tetraselmis convolutae* strain BS14 showed a dose-dependent response to cadmium, lead, and zinc. The toxic effect of the HMs exhibited the following decreasing order: Zn > Pb > Cd (Figure 2). The growth rates of *T. convolutae* exposed to 10 and 50 µM of Cd were only slightly lower than those of control cultures, while it was reduced to one-seventh at higher (e.g., 250 µM) concentrations. The exposure of *T. convolutae* to 50 µM of Pb reduced the cell density to almost half the density found in Pb-free cultures. As in the case of Cd contamination, 10 µM of Pb caused only a slight decrease in cell concentration, and at higher densities, a reduction to one-seventh of the cell density at higher (200 µM) doses. 

Compared to Pb and Cd, Zn was found to be more toxic to cells, causing a consistent inhibition (ca. 30%) of cell population growth, even in cultures amended with 20 µM of Zn. At higher doses of this metal, low (50 µM) or no (250 µM) microalgal growth was observed. Percentages of growth inhibition indicate low sensitivity of the *T. convolutae* strain BS14 towards cadmium and lead at concentrations below 50 µM, and clearly confirm that algal growth was seriously affected in Zn-amended media (Table 2). We calculated the growth inhibition based on the growth rate measured during the last 3 days of the experiments (i.e., between day 4 and day 7). The growth of *T. convolutae* strain BS14 was slightly limited at low and medium Cd concentrations and low Zn concentrations (Table 2). Low Pb concentrations did not inhibit microalgal growth. In contrast, microalgal growth was substantially inhibited at high Cd and Pb concentrations and intermediate Zn concentrations, whereas cells failed to grow at high Zn concentrations. 

### 3.3. Effect on Cell Morphology

We evaluated the impact of HM exposure on cell surface and morphology using light microscopy (Table 2, Figure 3). Cell surface area increased by up to 20% in the treatments containing the highest metal concentrations (Cd-250, Pb-200, and Zn-250). Specifically, *T. convolutae* strain BS14 exhibited a surface area of 55 ± 7.9 μm^2^ when cultured in ordinary f/2 medium, and this surface area increased to 66 ± 8 μm^2^, 64 ± 9.4 μm^2^, and 65 ± 9.4 μm^2^ with Cd, Pb, and Zn, respectively (Table 2). We observed some shape changes in specimens treated with both Cd and Zn (Figure 3); such variations might be related to alterations in the cell wall. In contrast, we did not observe substantial changes in Pb-treated specimens. Future studies using scanning electron microscopy and transmission electron microscopy are required to evaluate cell wall alterations due to HM exposure. Cell wall composition has been characterized in another strain of *T. convolutae* [59], as well as in *T. striata* [60], and mostly consists of acidic polysaccharides such as galactose and galacturonic acids. 

The results presented in Figure 3, showing *T. convolutae* cell behavior after 168 h in contact with Cd 250 µM, Pb 200 µM, and Zn 250 µM, are coherent with results described in Figure 2 and Table 1, and confirm the decreasing order of toxicity as the following: Zn > Pb > Cd (Figure 2).

### 3.4. HM Sorption in Algal Biomass

Cd biosorption was undetectable at low (i.e., 10 µM) concentrations, and varied between 0.23 and 0.38 µg mg^−1^ of dry weight at higher concentrations (50, 250 µM). Moreover, we did not observe a dose-dependent accumulation of this metal in algal biomass, since the results obtained at 50 and 250 µM were similar.

In Pb-treated cultures, a 10-fold increase in removal of this metal with respect to cultures exposed to the lowest dose (10 µM) was observed at higher Pb molarities (i.e., 50 µM). Cells adsorbed ca. 30% of this metal in 10 µM Pb-exposed cultures (Table 3); at higher doses (50 µM), adsorption was around 40%. We did not consider data regarding cultures exposed to 250 µM Pb, since aliquots of this metal precipitated in seawater consistently form complexes with certain f/2 nutrients without any interaction with algal cells, as already reported in previous studies [61]. 

Zn adsorption/compartmentalization was similar to that of the other metals (Table 3). Accumulation percentages varied among 40% at lower doses (20 µM), and ca. 24% at the highest tested dose (i.e., 250 µM). Cd, Pb, and Zn initial concentration and accumulation inside the cell biomass are shown in Table 3. 

## 4. Discussion

*Tetraselmis convolutae* has been previously observed in the Atlantic Ocean (GenBank accession numbers MT982710 and KT860916), suggesting that this ecotype has a broad geographic distribution, and can thrive in both pelagic and coastal waters. To the best of our knowledge, this is the first record of *T. convolutae* in the Tyrrenian Sea. The occurrence of *T. convolutae* in the Sarno River Mouth, one of the most polluted rivers in Europe, suggests that this species can tolerate a range of pollutants, including HMs. Indeed, species belonging to this genus have been previously observed in highly polluted environments, such as urban [62] or tannery wastewaters [63]. We chose to assess the potential of this test microorganism to Cd, Pb, and Zn, since these metals are commonly found in the Sarno River and its outfall [8], and to assess eventual differences in toxicity among essential (i.e., Zn) and non-essential (i.e., Pb and Cd) metals. 

Among them, Cd seemed to be the least toxic, causing a consistent decrease in cell density only at high (i.e., 250 µM) concentrations. Accordingly, Perez-Rama et al. [64] observed a strong cadmium tolerance exhibited by a congeneric species [64]. The relatively low toxicity of this metal has already been demonstrated in previous works performed with similar species, and may depend on the low or lack of internalization of this metal, which is mainly adsorbed on the cell surface [65,66], and/or on the ability of synthesizing metallothioneins as a possible mechanism of minimizing the detrimental effect of HMs [67]. ICP-MS analysis confirmed that low Cd concentrations were found in algal pellets at all doses, strengthening the hypothesis that, in our experiments, a low adsorption of this metal occurred. To better elucidate the weak interaction between Cd and *T. convolutae* cells, further investigations aimed at assessing the macromolecular composition of the cell wall of this specific genus, similar to those performed for *T. striata* [60], are mandatory.

A higher toxicity was observed in Pb-exposed cultures, where a strong reduction in cell growth already occurred in media amended with 50 µM of Pb, and a higher adsorption was observed relative to Cd-treated samples. These results are in contrast to those found for *T. chuii*, whereby Cd was three times more toxic than Pb [68]. On the other hand, Debelius and co-workers [69] found that *T. chuii* was able to adsorb Pb and to better tolerate it than other metals, such as copper (Cu), and the half maximal effective concentration (EC_50_) for Pb was 2–20-fold higher than those of other microalgal model species [69]. However, to our knowledge, only a few studies regarding the effect of Pb on the *Tetraselmis* genus are available in the literature. In general, the test organism seems to be more resistant to Pb than other strains belonging to the same genus, in which a reduction of 40% of the growth rate was observed even at low concentrations (ca. 0.1 µM) [70]. The higher tolerance of *T. convolutae* to Pb is likely due to a pre-adaptation of this specific strain to high concentrations of this metal in its natural environment [8]. Nonetheless, mechanisms of adsorption and eventual compartmentalization of Pb are still poorly investigated, so further experiments aimed at detecting and quantifying this metal in this genus are mandatory.

The current results disagree with a recent study by Kutlu and Dorucu [68], which highlighted a greater lethality of cadmium compared to zinc and lead on a congeneric species. These differences might be related to metabolic differences among the test microorganisms [68]. 

Zinc is an essential element for algal growth [42,71], and is thus included in almost all culture media at nanomolar concentrations [47,72]. The higher toxicity could be due to an easier intake of this element compared to the non-essential ones. However, while such information about the potential of microalgal biomass to remove metals from the aquatic environment is available in the literature, mechanisms highlighting Zn transport and internalization within microalgae are still poorly investigated. It has been suggested that zinc can enter the cell wall and be transported towards the vacuole in the diatom *Coscinodiscus eccentricus* [73]. A similar hypothesis was formulated by Santomauro and co-workers, who speculated that the coccolithophore *Emiliania huxleyi* can accumulate Zn in the cytoplasm after adsorption onto the cell wall, and is then able to remove it after sequestering it into vesicles [74]. The genus *Tetraselmis* is thus regarded as one of the most promising strains for Zn removal, since it seems more tolerant to this metal than both diatoms [75] and other green algae [76]. Preliminary observations under optical microscope highlighted some impairments in cell shape at high metal doses; however, we are aware that different techniques allowing a higher resolution of algal cells are mandatory to confirm the detrimental effects of HMs on this species. Eventual differences in algal cell size could be easily detected by flow cytometry cell sorting [77], and shape abnormalities with scanning or transmission microscopy [78]. In fact, new imaging techniques, such as tomographic phase microscopy, are available for a more accurate characterization of microalgal cells [79]. Indeed, they allow the detection of the eventual alteration in both cell and organelle morphology, and could be suitable to detect the main detrimental effects experienced by *T. convolutae* at the cellular and subcellular level.

In summary, we assessed the potential of an autochthonous strain to remove HMs from seawater. These preliminary results seem to confirm that this strain could be suitable for future applications aimed at reducing HM concentrations in estuarine and seawater environments, since it is generally able to tolerate concentrations of at least 10 µM of all the tested metals without dramatic reductions in growth rates and/or impairments in cell structure and morphology. It is worth noting that the tested concentrations are all higher than those usually found in polluted areas [8]. On the other hand, the synergistic effects of different pollutants and longer periods of exposure to HMs are usually experienced by microalgae in the natural environment; thus, further experiments aimed at investigating the concomitant effect of more metals on this species are mandatory to assess its potential for HM bioremediation purposes. Moreover, further investigation into the molecular mechanisms involved in the HM response in *T. convolutae* is mandatory to better assess the potential of this species towards the decontamination of certain metals. Previous studies have shown that a similar species (*T. suecica*) is able to synthesize class III metallothioneins [66] and phytochelatins [79], which allow a strong tolerance towards Cd; however, to our knowledge, information about other metals, such as Pb and Zn, is still scarce. Finally, since *Tetraselmis* species can be easily cultured in large volumes in both open and enclosed systems [80,81,82,83,84,85], experiments on preindustrial scales (i.e., with high working volumes) should be performed to predict the potential of the test organism for ex situ or in situ strategies for bioremediation of HM-polluted environments.

## 5. Conclusions

In this work, we identified *Tetraselmis convolutae* in the Gulf of Naples for the first time, and tested its resistance to both essential (Zn) and non-essential (Pb, Cd) metals. Our results confirm that this microorganism is able to tolerate high HM concentrations, and could thus be employed as a biosorbent in HM-polluted environments. However, further studies are mandatory to test its resistance to other metals as well as possible synergistic effects on their growth and adsorption rates. Detrimental effects on algal shape at very high HM doses seem to confirm their potential as sentinel species to detect phenomena of acute HM pollution. Further investigation into the impairments in algal and organelle morphology with high-resolution imaging techniques is mandatory to confirm their potential use as sentinel species to detect HM pollution.

## Figures and Tables

**Figure 1 microorganisms-10-02445-f001:**
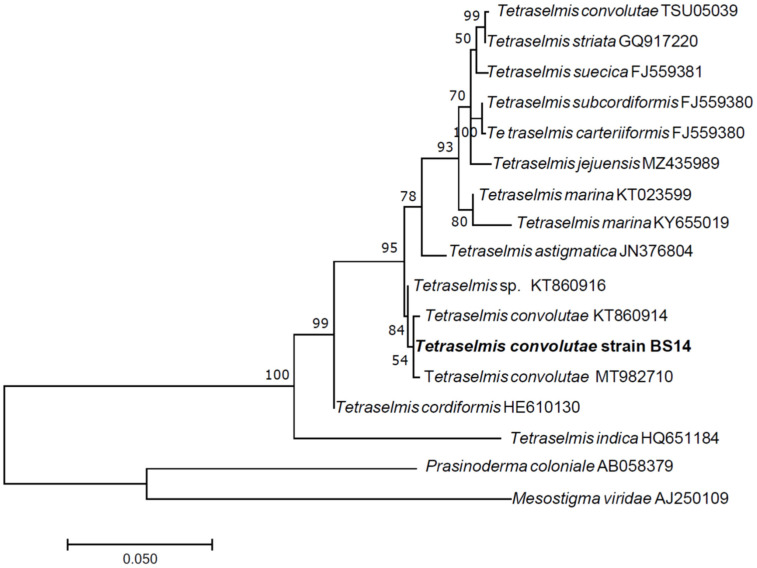
Partial 18S rRNA gene phylogenetic tree of *Tetraselmis convolutae* strain BS14. The evolutionary history was inferred using the maximum likelihood method based on the Kimura 2-parameter model [57]. A discrete Gamma distribution was used to model evolutionary rate differences among sites. The percentage of trees in which the associated taxa clustered together is shown next to the branches. The tree is drawn to scale, with branch lengths measured in the number of substitutions per site. The analysis involved 17 nucleotide sequences. There were a total of 1105 positions in the final dataset. Evolutionary analyses were conducted in MEGA7 [58].

**Figure 2 microorganisms-10-02445-f002:**
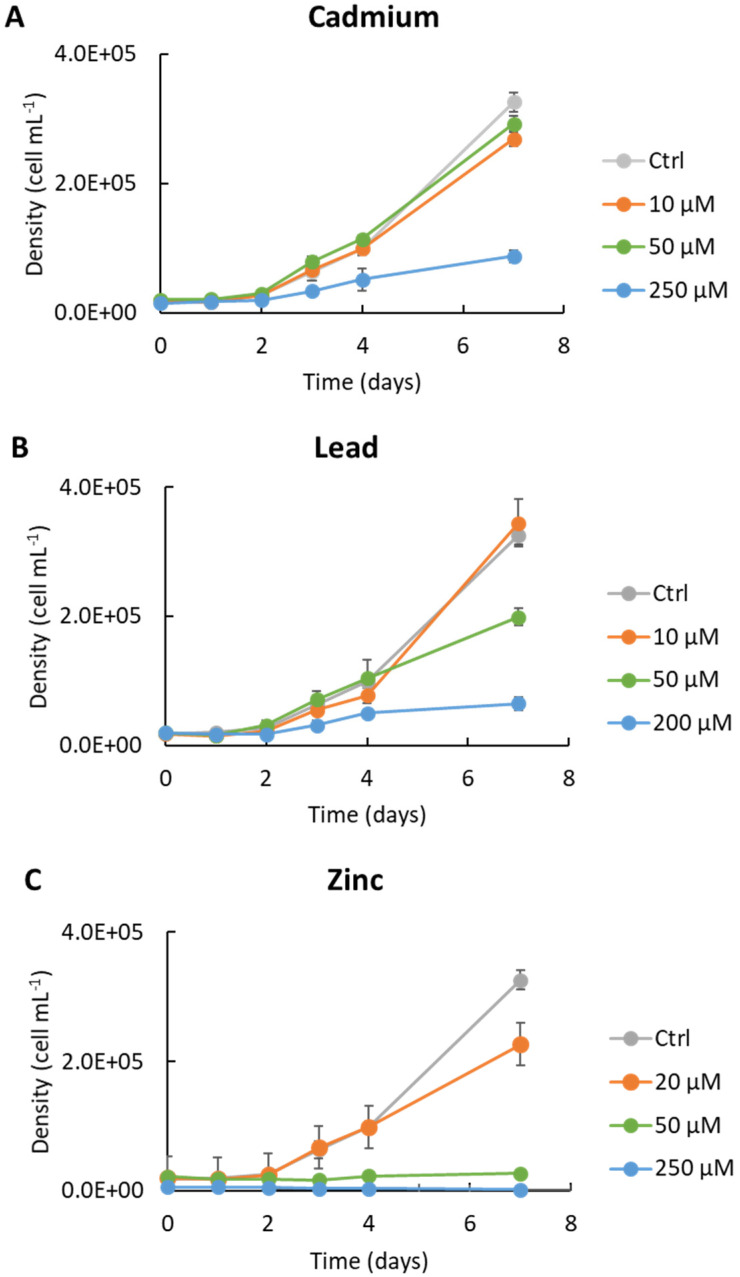
Effects of exposure of Cd (**A**), Pb (**B**), and Zn (**C**) on cell density. Microalgae were grown in liquid f/2 medium containing heavy metals at scalar doses. Each experiment was performed in triplicate; dark grey bars indicate standard deviations.

**Figure 3 microorganisms-10-02445-f003:**
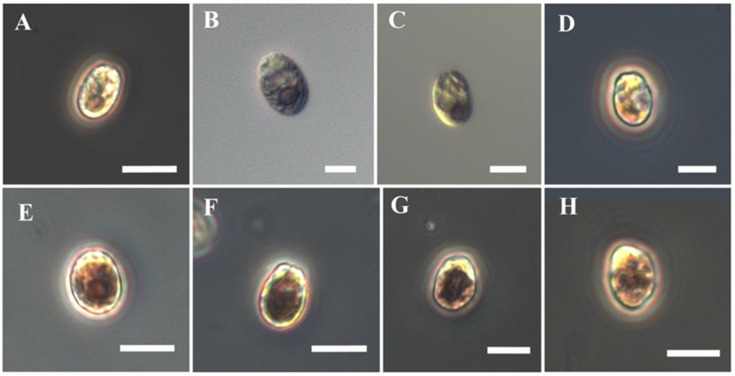
Phase contrast (**A**,**D**,**E**–**H**) and differential interference contrast (**B**,**C**) micrographs of *Tetraselmis convolutae* strain BS14 cultured in the presence of different HMs for 168 h: (**A**,**B**) cells cultured in EDTA-free f/2 without HMs; (**C**,**D**) cells treated with 250 µM Cd; (**E**,**F**) 200 µM Pb; (**G**,**H**) 250 µM Zn. Scale bars: (**A**,**E**–**H**) = 10 µm, (**B**–**D**) = 5 µm.

**Table 1 microorganisms-10-02445-t001:** Concentration of heavy metals added to the culture medium for the experiments (* three replicate flasks were used for each treatment). Numbers next to the treatment IDs correspond to the micromolar concentration of the corresponding metal. Cd, Pb, and Zn salts used for the experiments are indicated below the name of the corresponding metal.

Treatment *	Metal	Concentration (mg/L)
Ctrl	-	-
Cd-10	Cadmium (as CdCl_2_)	1.1
Cd-50		5.6
Cd-250		28.1
Pb-10	Lead (as PbNO_3_)	2.1
Pb-50		10.4
Pb-200		22.4
Zn-20	Zinc (as ZnSO_4_·7H_2_O)	1.3
Zn-50		3.3
Zn-250		16.4

**Table 2 microorganisms-10-02445-t002:** *Tetraselmis convolutae* inhibition response in presence of scalar doses of Pb, Cd, and Zn (statistical analysis was performed with post hoc ANOVA and Dunnett’s test with *p* < 0.05; cell surface values are expressed as mean ± standard deviation). Cell surface was calculated only at higher doses that caused visible changes under the optical microscope (n.c., not calculated).

HM	Growth Rate Inhibition (% of Control)	Cell Surface (µm^2^)
Ctrl	-	55 ± 7.9
Cd-10	7.6 ± 2.9	n.c.
Cd-50	6.7 ± 1.5	n.c.
Cd-250	39 ± 3	66 ± 8
Pb-10	0	n.c.
Pb-50	20 ± 1.8	n.c.
Pb-200	59 ± 4.7	64 ± 9.4
Zn-20	16 ± 2.4	n.c.
Zn-100	93 ± 1.3	n.c.
Zn-250	100 ± 3.7	65 ± 9.4

**Table 3 microorganisms-10-02445-t003:** Concentration of HMs measured by ICP-MS in both the medium and microalgal biomass. Results are expressed as mean ± standard deviation.

HM Typology and Molar Concentration (µM) (Theoretical Concentration)	Real HM Concentration Measured by ICP-MS (µM)	µg/mg	pg/cell
Pb 10 µM	11.58	1.19 ± 0.06	2.04 ± 0.30
Pb 50 µM	47.78	9.77 ± 0.91	28.26 ± 2.00
Cd 50 µM	47.15	0.3 ± 0.00	0.67 ± 0.07
Cd 250 µM	226.85	0.31 ± 0.10	1.97 ± 0.11
Zn 20 µM	23.71	0.83 ± 0.03	2.56 ± 0.49

## Data Availability

Not applicable.

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
