# Peer review of "Identification of a Green Algal Strain Collected from the Sarno River Mouth (Gulf of Naples, Italy) and Its Exploitation for Heavy Metal Remediation"

_microorganisms, 2022, doi:10.3390/microorganisms10122445_

Round 1

Reviewer 1 Report

Submitted manuscript titled “ Identification of a green algal strain collected from the Sarno River mouth and its exploitation for heavy metal remediation” addresses a research study  determining the ability of an endemic microalgal strain, Tetraselmis convolutae BS14, to grow in polluted seawater containing different concentrations of cadmium, lead or zinc.

The work is well-planned and structured. The methodology is well described and used correctly, and the number of data is very high. The results are clearly displayed, and the quality of the figures is good. Seventy-eight references were cited in this manuscript, most of which were published in the past five years, suggesting that the paper's results are new and have not been extensively researched in recent years. In my opinion, the research could be interesting for a broad audience and has the potential to be published. However, some issues should be considered before possible publication. General comments and suggestions for the authors are as follows:

1. Content should be centred in table 1.

2. In what form were the metals added? (line 158).

3. Website in line 132 should be cited in the references.

4. The growth inhibition formula (line 169), expressed as a percentage, should be multiplied by 100%.

5. The analytical methods for determining heavy metals need a more detailed description concerning the analytical metrics: uncertainty, limits of detection, selectivity etc.

Author Response

  1. Content should be centred in table 1.

done

  1. In what form were the metals added? (line 158).

We had already inserted in Table 1 the chemical formula of the metallic salts used for the experiments. However, we agree with the referee that this information should be also clarified in the table caption, that has been modified according to his/her suggestion

  1. Website in line 132 should be cited in the references.

We thank the referee for this suggestion. In order to be even more precise on this aspect, we inserted the as reference the article of dr. Hall (1999), in which the operation of the programme had been described for the first time

  1. The growth inhibition formula (line 169), expressed as a percentage, should be multiplied by 100%.

We corrected the formula in the text of the manuscript

  1. The analytical methods for determining heavy metals need a more detailed description concerning the analytical metrics: uncertainty, limits of detection, selectivity etc.

We have added some extra info in the material and methods section (detection limits and linear range of the calibration curve). We have also added the standard deviation of the measurements in table 3.

Reviewer 2 Report

The MS entitled "Identification of a green algal strain collected from the Sarno River mouth and its exploitation for heavy metal remediation" is very interesting. The work is well conducted and they have performed a good study. Although it is potentially interesting, minor revisions are necessary.

1) Please add the novelty in Abstract.

2) Please add the hypothesis at the end of the introduction.

3) What is the meaning of ca in line no. 92. Authors should write their full name when mentioned for the first time.

4) Discussion needs more citations.

5) Where is the static analysis in tables and figures.

6) I strongly suggest designing the graphical abstract in your MS.

Author Response

1) Please add the novelty in Abstract.

We thank the referee for this important suggestion, we have emphasized the innovative aspects of our research; indeed, it is the first time that a novel microalgal strain collected from the Sarno River mouth has been employed for HM removal from aquatic environments. Moreover, our results demonstrate that this new isolate can be a suitable candidate for bioremediation purposes. We did our best to synthesize this information, since, according to authors’ guidelines, the abstract should be a total of about 200 words maximum. However, we reiterated these concepts within the text of the manuscript.

2) Please add the hypothesis at the end of the introduction.

Done. We stated that our experimental results suggest that this species could be considered a good candidate for bioremediation of HM-polluted waters, and that its efficiency is particularly evident towards the non-essential metals Pb and Cd.

3) What is the meaning of ca in line no. 92. Authors should write their full name when mentioned for the first time.

The abbreviation “ca.” stands for “circa”, that means “approximately”. The did not use the extended word since this abbreviation is commonly used in both common and scientific written English. However, if the referee finds it inappropriate, we can use the term “approximately”.

4) Discussion needs more citations.

We are aware that we have few reference studies to make comparisons among our results and results obtained from other research studies, but we can ensure that we did a very accurate bibliography research that was focused on the effect of HMs on the genus Tetraselmis. Our research included the occurrence of this genus in polluted waters, detoxification strategies, capabilities of HM adsorption/accumulation, as well as HM pollution in the collection site (i.e. the Sarno River mouth). Unfortunately, literature regarding the main topics of the present paper is still scarce, but we consider this aspect a strong point of our work, since it has several innovative aspects (the occurrence of the species T. convolutae for the first time in the Sarno River, an accurate study of HM accumulation on a novel strain, its high tolerance to non-essential metals such as Pb and Cd).

However, we added more references related to: 1) the effects of Cd on the genus Tetraselmis (we did not find additional information regarding the other metals) and 2) the ability of this species to growth at industrial scale (we declared that follow-up studies on larger volumes are mandatory, but we did not insert in the previous version the related literature).

5) Where is the static analysis in tables and figures.

We inserted this information in the table and figure caption. Results are usually expressed as mean ± standard deviation, and we clarified it in the new version.

6) I strongly suggest designing the graphical abstract in your MS.

We have designed a graphical abstract that represents the main experimental activities described in the present paper, from sample collection to algal identification and experiments of HM exposure

Round 2

Reviewer 1 Report

My recommendation for the article "Identification of a green algal strain collected from the Sarno River mouth and its exploitation for heavy metal remediation" - accept in its present form.